# A tool for determining multiscale bedform characteristics from bed elevation data

Judith Y. Zomer[1], Suleyman Naqshband[1], and Antonius J. F. Hoitink[1]

[1]Department of Environmental Sciences, Hydrology and Quantitative Water Management Group, Wageningen University & Research, Wageningen, Netherlands

**Correspondence:** Judith Zomer (judith.zomer@wur.nl)

**Abstract.** Systematic identification and characterization of bedforms from bathymetric data are crucial in many studies of fluvial processes. Automated and accurate processing of bed elevation data is challenging where dune fields are complex, irregular and, especially, where multiple scales co-exist. Here, we introduce a new tool to quantify dune properties from bathymetric data representing large primary and smaller, superimposed, secondary dunes. A first step in the procedure is to decompose the bathymetric data using a LOESS algorithm. Steep lee side slopes of primary dunes are preserved by implementing objective breaks in the algorithm, accounting for discontinuities in the bed elevation profiles at the toe of the lee side slope. The steep lee slopes are then approximated by fitting a sigmoid function. Following the decomposition of the bathymetric data, bedforms are identified based on zero-crossing, and morphological properties are calculated. The approach to bedform decomposition presented herein is particularly applicable where secondary dunes are large and thus filtering using conventional continuously differentiable functions could easily lead to undesired smoothing of the primary morphology. Application of the tool to two bathymetric maps demonstrates that it successfully decomposes bathymetric data, identifies primary and secondary dunes, and preserves steeper lee side slopes of primary dunes.

## 1 Introduction

Dunes are rhythmic features that develop at the interface of a flow field and a mobile bed. In fluvial environments, dunes play an important role in various flow and transport processes, on multiple scales. Flow separation downstream of steep dunes opposes the mean flow, increasing hydraulic roughness (Maddux et al., 2003b, a). Moreover, turbulent flow structures generated over dunes play a key role in the generation of instantaneous bed shear stresses and bedload sediment movement and resuspension (Nelson et al., 1993, 1995; McLean et al., 1999; Cellino and Graf, 2000; Bradley et al., 2013). In river management, bedform dynamics are of interest for fairway navigability, flood risk protection and stability of infrastructure. As a result, over the past decades, fluvial dunes have been a subject of extensive research.

Systematic identification and characterization of dunes from bed elevation scans greatly aid these research efforts. Examples are field and flume studies which investigate the development of dunes under a range of conditions (Bradley and Venditti, 2017; Reesink et al., 2018; Venditti et al., 2016; Cisneros et al., 2020; Naqshband and Hoitink, 2020; Wilbers and Ten Brinke, 2003; Chen et al., 2012). Understanding and predicting the relationship between hydrodynamics and dune characteristics is of vital

importance for operational modelling and flood risk assessments. Similarly, field and flume studies aim to discover how dune characteristics impact the mean and turbulent flow field, and the associated sediment dynamics (Kwoll et al., 2016; Lefebvre et al., 2016; Parsons et al., 2005; Best and Kostaschuk, 2002; Bradley et al., 2013). Based on dune shape and migration speed, observed in the field, bedload sediment transport can be quantified (Abraham et al., 2011; McElroy and Mohrig, 2009; Simons et al., 1965). Each of these research fields requires dune identification and characterization, which are particularly challenging where dunes fields are complex, irregular and where multiple scales co-exist.

In fluvial systems, two scales of dunes often co-exist: larger, primary dunes and small, secondary dunes that are superimposed on the primary dunes. The secondary bedforms have long been considered an attribute of primary dunes, converting simple dunes into compound dunes (Ashley, 1990). Recent studies have shed light on the relevance of the small bedforms, which have been observed in river systems worldwide (Carling et al., 2000; Cisneros et al., 2020; Galeazzi et al., 2018; Harbor, 1998; Parsons et al., 2005; Wilbers and Ten Brinke, 2003; Zomer et al., 2021). Secondary bedforms are not limited to the primary dune stoss, but can migrate over of the full length of the primary bedform (Galeazzi et al., 2018; Zomer et al., 2021). They possess steep lee side angles which are likely to influence the total roughness, and to affect primary dune development as well (Reesink and Bridge, 2007). The bedload transport associated with secondary bedform migration is similar to that associated with primary dunes, due to their high migration speed (Zomer et al., 2021; Venditti et al., 2005). The expected increase of spatial and temporal resolution of bathymetric data sets further stimulates research on this smallest dune scale.

Various methods to quantify dune morphology have been developed (van Dijk et al., 2008; Gutierrez et al., 2013; Cisneros et al., 2020; Scheiber et al., 2021; Van der Mark and Blom, 2007; Wang et al., 2020; Lefebvre et al., 2021). In dealing with multiple scales of bedforms, several methods isolate bedforms based on size (Cisneros et al., 2020; Scheiber et al., 2021). This may not be suitable where the primary dune scale is fully covered by secondary dunes (Galeazzi et al., 2018; Zomer et al., 2021). Other methods apply filtering based on geostatistics (van Dijk et al., 2008), spectral methods (van Dijk et al., 2008; Cazenave et al., 2013; Lee et al., 2021) or spline functions following spectral analysis (Gutierrez et al., 2013; Wang et al., 2020). When using spectral filters, the reconstruction of the asymmetric shapes of high-angle dunes cannot do without high-frequency base functions, which are removed in the filtering process, in turn leading to a misrepresentation of the steep lee side slope in the lowpass filtered signal (Lee et al., 2021). Also in other filtering methods, a high degree of smoothing, which is inevitable in the presence of larger secondary bedforms, significantly affects the primary lee side slope, leading to missing secondary bedforms on the lee side, or at the least significant underestimation of the lee side slopes. Smoothing filter might also distort the morphology of secondary bedforms if the highpass filtered signal used for the characterisiation of secondary bedforms.

In this study we develop a new tool to quantify bedform characteristics from bathymetric data representing multiple scales. The initial bedform identification is based on zero-crossing after decomposition of the bathymetric data. A LOESS algorithm is used to fit the irregular larger scale morphology, including the primary dunes. LOESS regression is a nonparametric technique that uses local weighted regression to fit a smooth curve through points in a scatter plot. The approach differs from previous methods (van Dijk et al., 2008; Cazenave et al., 2013; Gutierrez et al., 2013; Wang et al., 2020; Lee et al., 2021), in that no continuously differentiable function is fitted, but rather breaks are implemented. These breaks in the LOESS fit are used to

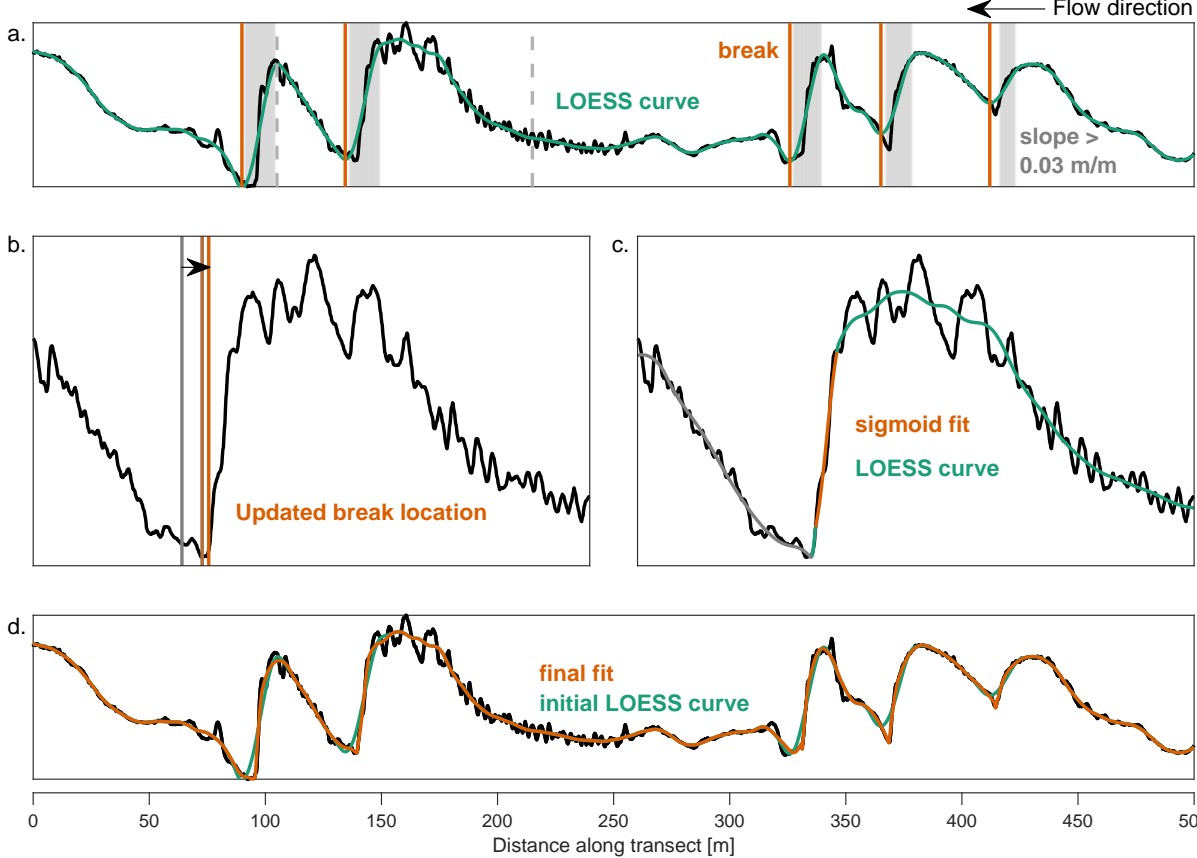

**Figure 1.** Schematic overview of the tool. a: Bed elevation series with the initial LOESS-based smoothed curve. The gray areas indicate where the lee slope in the LOESS curve exceeds $0.03\,\mathrm{m\,m^{-1}}$, and the corresponding breaks are indicated with vertical lines. b: Break locations after updating. c: A sigmoid function is fitted to the lee side slope. d: The bathymetric signal with the initial LOESS curve and final result.

avoid smoothing of steep primary lee side slopes that contain an abrupt transition from the lee side slope to the dune trough. These steep lee slopes are approximated with a sigmoid function. The tool is applied to two bathymetric data sets to illustrate the method.

## 2 Methods

The first step in the procedure is to decompose bed elevation profiles (BEPs) into a signal representing secondary bedforms
and the remainder, which includes the primary dunes. This step builds on the previously described method to decompose bed elevation profiles by Zomer et al. (2021). The decomposition is described in section 2.1. Subsequently, secondary and primary

bedforms are identified based on zero-crossing, which is described in section 2.2. In Section 2.3, the new tool is applied to two data sets.

## 2.1 Decomposition of bed elevation data representing multiple bedform scales

A LOESS curve is fitted to the data to separate the secondary bedforms from the underlying morphology (Greenslade et al., 1997; Schlax and Chelton, 1992). At steep primary leesides, with slopes exceeding a user-defined threshold, breaks are introduced and a sigmoid function is fitted to the corresponding primary lee side using a least-squares routing. First, the LOESS algorithm and sigmoid function are explained in further detail, then the subsequent steps in the method are provided.

By fitting the LOESS curve, each optimized value is given by a weighted quadratic least squares regression to a local subset
of the data. So, for each grid point in a bed elevation profile $(x_0)$, a local estimate $\hat{z}$ is found through a locally weighted least squares fit of a function of $x$, applied to $N$ data points near $x_0$ (Greenslade et al., 1997):

$$\hat{z} = a_1 + a_2 x + a_3 x^2. \tag{1}$$

The coefficients $a_1$, $a_2$, and $a_3$ are found minimizing the function

$$\Phi = \frac{1}{W} \sum_{j=1}^{N} w_j^2 (\hat{z} - z)^2, \tag{2}$$

where W is the sum of weights $w_j$. The weights $w_j$ are defined by a tricube weight function as

$$w_j = \begin{cases} (1 - q_j^3)^3, & 0 \le q_j \ge 1 \\ 0 & q_j < 1 \end{cases} \tag{3}$$

$$q_j = \left( \frac{x_j - x_0}{d_x} \right)^2, \tag{4}$$

where $d_x$ is the half-span of the smoother.
The sigmoid function that is fitted to dune lee sides is defined as

$$\hat{z} = b_1 + \frac{b_2}{1 + e^{-b_3(x - b_4)}}. \tag{5}$$

The coefficients are found by minimizing

$$\Phi = \sum_{i=1}^{N} (\hat{z}_i - z_i)^2 \tag{6}$$

through constrained nonlinear optimization using an interior-point method in MATLAB implemented as "fmincon". A user
may choose to use an alternative smoothing algorithm to replace LOESS, or use an alternative S-shaped function to replace the sigmoid function. Examples of smoothing algorithms are the Savitzky-Golay filter or a kernel smoothing algorithm. The sigmoid curve could potentially be replaced by a hyperbolic tangent function. The LOESS algorithm was selected here because

it is considered appropriate to fit primary dunes with irregular shapes in terms of heights and lengths and where deformation of bedforms is significant (Ganti et al., 2013). Also, no information is lost at the start and at the end of a spatial series. Practical considerations in selection of an algorithms could be computational time and whether data points are equidistant or not.

The bed elevation data series is decomposed based on both the LOESS curve and fitting of the steep primary lee sides with a sigmoid function fit. Data input are (curvilinear) grids. The methodology is applied per bed elevation profile (BEP). The subsequent steps are:

1. An initial LOESS curve is fitted to the BEP (Figure 1a).

2. Based on the initial LOESS curve, crest and trough locations of primary dunes are identified. If the maximum lee side slope in the LOESS curve is larger than a specified value (default: $0.03\,\mathrm{m\,m^{-1}}$, which is approximately $1.7°$), a break is set at the corresponding trough. If there are no breaks in a BEP, the initial LOESS curve is retained (Figure 1a).

3. If there are breaks in a BEP, the exact locations of these breaks are updated in the following steps. First, the local minimum of the bed elevation data is found, within a specified window upstream of the previous breakpoint. Subsequently, the breakpoint location is updated if the slope to an upstream location within the window is lower than the cutoff slope (Figure 1b).

4. A LOESS curve is fitted up to the first break. Then, at the primary lee side upstream of the break, the sigmoid function is fitted, with initial values $b_1 = z_{break}$ and $b_4 = x_{break}$. Only the central section of the sigmoid function is retained for the eventual decomposition: the values with a slope smaller than 0.5 times the maximum slope or smaller than the cutoff value (default $0.03\,\mathrm{m\,m^{-1}}$) are removed. Also, if part of the fit is lower than the local minimum near the trough, this part is removed (Figure 1c).

5. A short LOESS curve is fitted to the data between the previous LOESS curve and the sigmoid function fit. The fit is forced to connect to LOESS curve and sigmoid fit by artificially adding data points (Figure 1c).

6. Steps 4 and 5 are repeated (Figure 1d).

## 2.2 Bedform identification and characterization

The identification of both primary and secondary bedforms is based on the decomposed bed elevation signals. The characterization of bedforms includes the following properties: height, length, depth, trough and crest locations, lee side slope, maximum lee side slope, stoss side slope, and the aspect ratio.

Secondary bedform identification is based on zero-crossing applied to the decomposed bed elevation signal ($z - z_{loess}$), following Van der Mark and Blom (2007). A zero-crossing is marked as a downcrossing if the slope is negative, and as upcrossing if the slope is positive. The crests and troughs are determined as local maxima and minima between up- and downcrossings. For the secondary bedforms, we apply one iteration in the bedform identification, in order to eliminate very small fluctuations around the zero-line. Secondary bedform properties are determined both for the unfiltered bathymetric data (based

on previously identified crest and trough locations) and for the decomposed bathymetric data. The bedform height is defined as $z_{crest} - (z_{trough-upstream} + z_{trough-downstream})/2$. The length is defined as the horizontal distance along the (curvilinear) grid between the up- and downstream trough. The depth per bedform is computed as the average bed level (based on unfiltered bathymetric data) between the up- and downstream trough. The lee slope is defined as the average slope between the crest and downstream trough. The maximum lee slope is the maximum slope of a grid cell between the crest and downstream trough. The stoss slope is characterized as the average slope between the crest and upstream trough. The aspect ratio is the height divided by length.

The identification and characterization of primary bedforms is similar to that of secondary bedforms. Zero-crossing is based on the decomposed signal, excluding secondary bedforms, and a base level. The latter can be computed as a moving average (4 to 5 times the dune length), a smoothed LOESS curve or a time-average of the local river bed, depending on data availability. Here, we compute the base level as a moving average. Primary dunes are identified, iterating once. If a primary dune height is smaller than 0.25 m, corresponding up- and downcrossings are removed and new minima and maxima are found. Properties are subsequently determined similar to secondary bedforms. The maximum lee slope is defined as the maximum slope of a grid cell between the crest and the downstream trough. If the primary lee side is fitted using the sigmoid function, the maximum slope is determined based on values corresponding to the fitted function only.

After bedform identification and characterization, the secondary and primary bedforms are filtered to exclude bedforms that are deemed unrealistic, such as small fluctuations around the zero-line. Secondary bedforms are filtered out if one or more of the following conditions hold, which are all user-defined and site-specific:

- height is smaller than 0.05 m or larger than 0.75 m

- length is larger than 25 m or smaller than 0.5 m (5 times the resolution for data presented herein)

- the aspect ratio is larger than 0.2 or smaller than 0.005

- the crest elevation in the unfiltered data is less than 0.01 m lower than the up- or downstream trough

- the maximum lee side slope is smaller than $0.03 \, \mathrm{m\,m^{-1}}$.

Primary bedforms are filtered out if:

- height is smaller than 0.25 m or larger than 4 m

- length is larger than 200 m or smaller than 25 m

- aspect ratio is larger than 0.2 or smaller than 0.005

- the maximum lee side slope is smaller than $0.03 \, \mathrm{m\,m^{-1}}$.

All default values in the procedure above can readily be adjusted in the code.

## 2.3 Data description

The tool is applied to two data sets. Multibeam echosounding (MBES) data were provided by the Dutch Ministry of Infrastructure and Environment (Rijkswaterstaat) for the River Waal, which is the main branch of the River Rhine. The used data set consists of one kilometer of the river, from approximately 425370 N, 154178 E to 426252 N, 154618 E (EPSG:28992). The first data set was acquired on 25 August 2017, when the discharge in the Rhine at Tiel, a nearby station, was 1204 $\text{m}^3\text{s}^{-1}$. The second data set was acquired on 23 January 2018, when the discharge was 3647 $\text{m}^3\text{s}^{-1}$. The data were provided as point clouds and were interpolated through inverse distance weighting on a curvilinear grid with an approximate longitudinal horizontal resolution of 0.1 m, an approximate lateral resolution of 1 m and a vertical resolution of 0.01 m. BEPs analyzed here include every longitudinal grid line between -81 m and 82 m with respect to the river's central axis, which is the region unaffected by scours induced by river groynes.

## 3 Results

The two data sets were decomposed with the following parameters: [$d_x = 14$, window = 12, cutoff slope = 0.032] for the data set acquired on 25 August 2017 and parameters [$d_x = 21$, window = 16, cutoff slope = 0.03] for the data set acquired on 23 January 2018. The bathymetric maps, primary bed elevation series and the signal corresponding to secondary bedforms are shown in Figure 2. The campaign in August followed an extended period of low discharge. Secondary bedforms were relatively small and only partly cover the primary dunes. The campaign in January 2018 took place during the rising limb just before a peak discharge, which succeeded several similar peaks in the previous months. During high discharges, primary dunes are shorter, and large secondary bedforms cover a large part of bed. In the southern river section secondary bedforms are dominant whereas primary dunes nearly disappear. In the two cases presented here, secondary bedforms are not part of of a decaying process, where they cannibalize larger primary dunes (Bradley and Venditti, 2021). Two scales coexist during low flow and a rising hydrographic limb, similar to what has been observed in the Waal river by Zomer et al. (2021).

Following the decomposition of the bed elevation data, bedforms have been identified based on zero-crossing. For primary bedforms, the base level consists of a moving average of 400 m. Four BEPs are shown in Figure 3, which shows for each BEP the measured bed elevation, the fitted line, and the locations of the crests and troughs of secondary and primary bedforms. Figure 4 shows the distributions of height, length and the maximum lee side slope that are found for primary and secondary bedforms based on the unfiltered bathymetric data.

In the applied decomposition procedure, breaks were implemented to avoid smoothing of steep primary lee side slopes. To investigate the effectiveness of the procedure, histograms of the slopes of downstream facing cells with a value larger than 0.03 $\text{m m}^{-1}$ are shown in Figure 5. The first column shows the distribution of lee side slopes of the original bed elevation data. The second column shows the same for the primary morphology after decomposition without breaks. Here, LOESS curves are fitted with the parameters as reported in the Results section. The third column shows the results for the primary morphology using the sigmoid function fit to the slopes. The figure indicates that introducing a break in the BEP and a separate treatment of the lee side slopes increases the tail of the distribution towards the higher slopes.

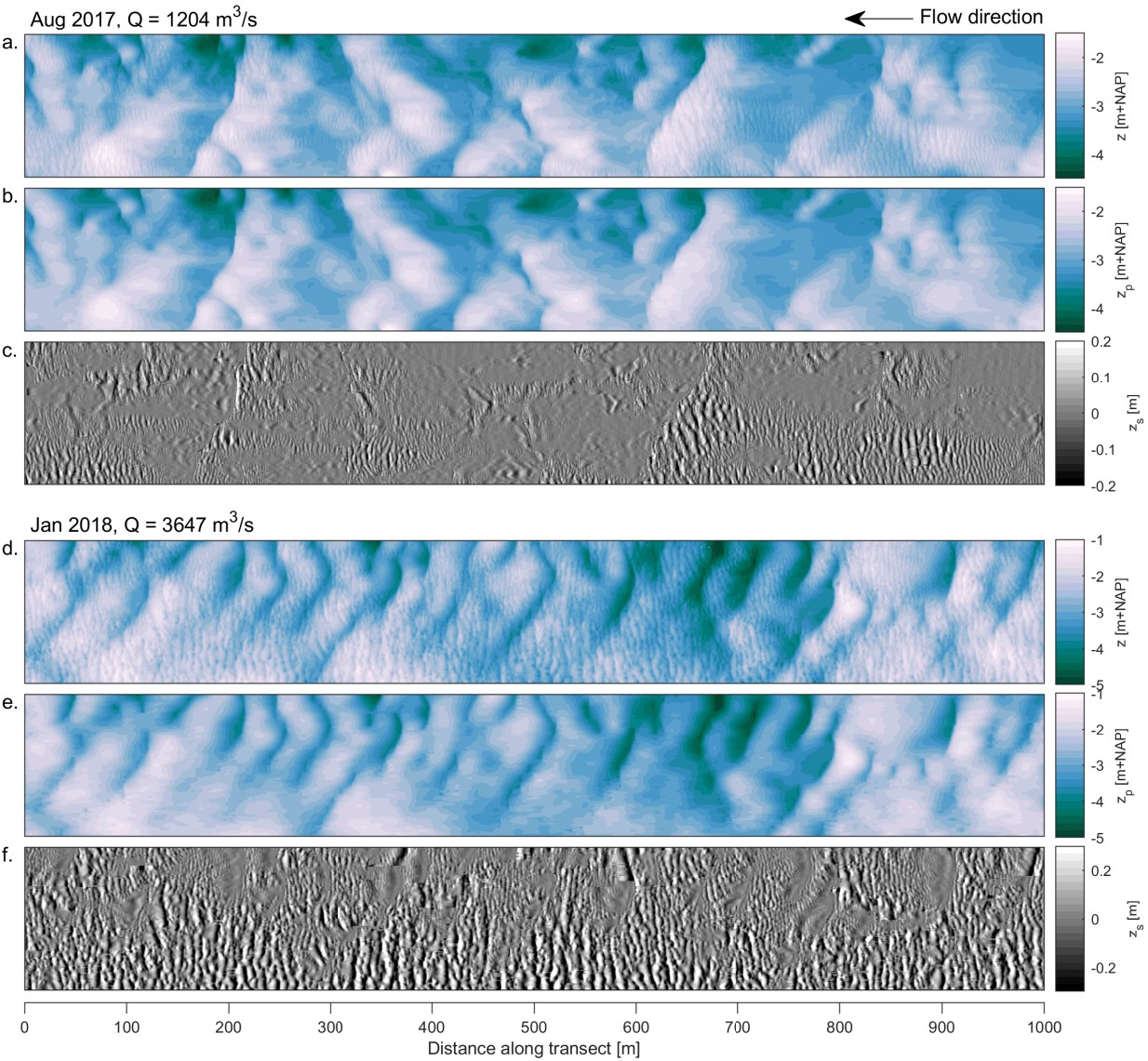

**Figure 2.** The bathymetric maps before and after decomposition. a, d: the initial bathymetry. b, e: the primary morphology. c, f: the secondary bedforms.

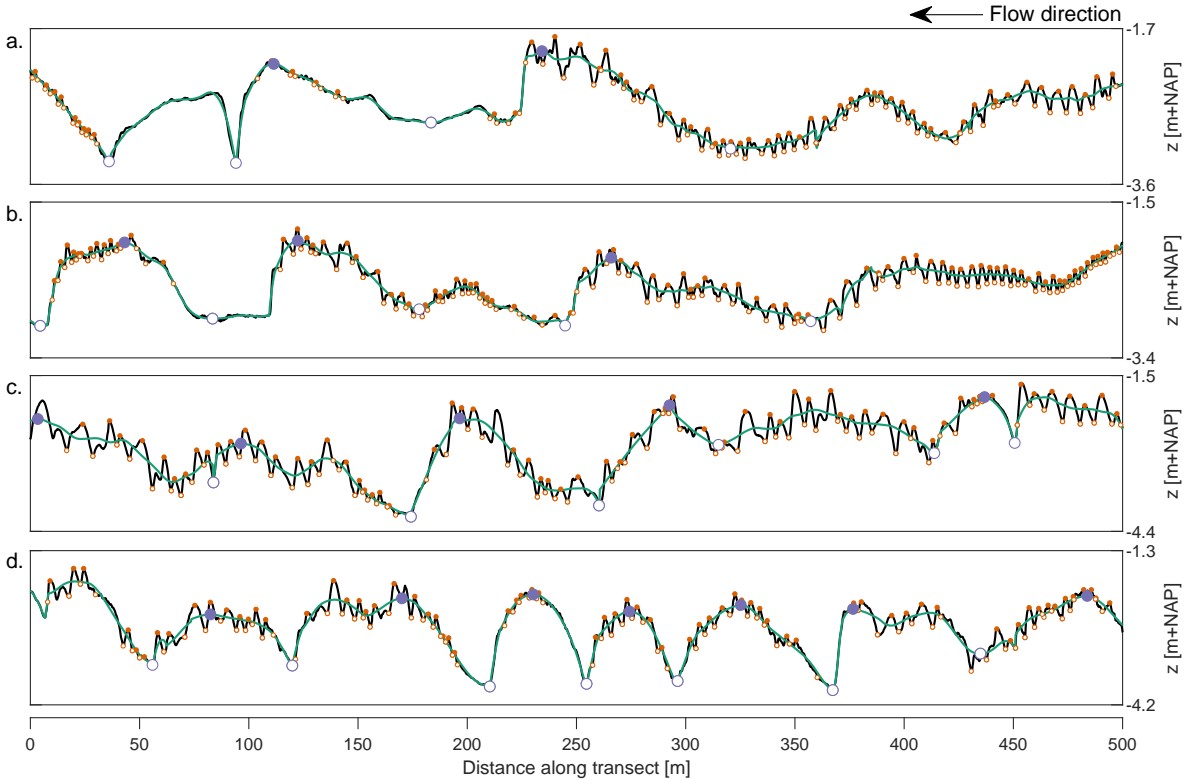

**Figure 3.** Four example BEPs. Each panel shows the bed elevation signal, the fitted line and the crests and troughs of the primary (large, purple circles) and secondary dunes (small, orange circles). a, b: BEPs from the data set acquired in August 2017. c, d: BEPs from the data set acquired in January 2018.

## 4    Discussion

The new tool presented herein serves two purposes: to isolate secondary bedforms from the underlying topography, and to identify bedform properties for both the primary and secondary bedforms based on zero-crossing. Figure 2 and 3 demonstrate that the tool decomposes the bathymetry well. The secondary bedforms are separated from the underlying bed topography and steep lee side slopes maintain their steepness. High-pass filtered data, shown in Figure 2c and f, indicate that the primary dune shape is not present in this signal, whereas in Figure 2b and e, no indication of secondary bedforms is present, indicating they are filtered out effectively. Whereas previous methods employ smoothing functions that have a continuous first derivative, such an approach is unsuitable for bathymetries with relatively large secondary bedforms, because a high degree of smoothing fails to adequately represent dune lee side slopes. For this reason, breaks are implemented here, leading to a fitted line that is continuous, but does not have a continuous first derivative. Figure 5 shows that by implementing breaks, the steep primary lee side slopes are not smoothed during the decomposition. Accurate estimation of lee side slopes is of vital importance because lee

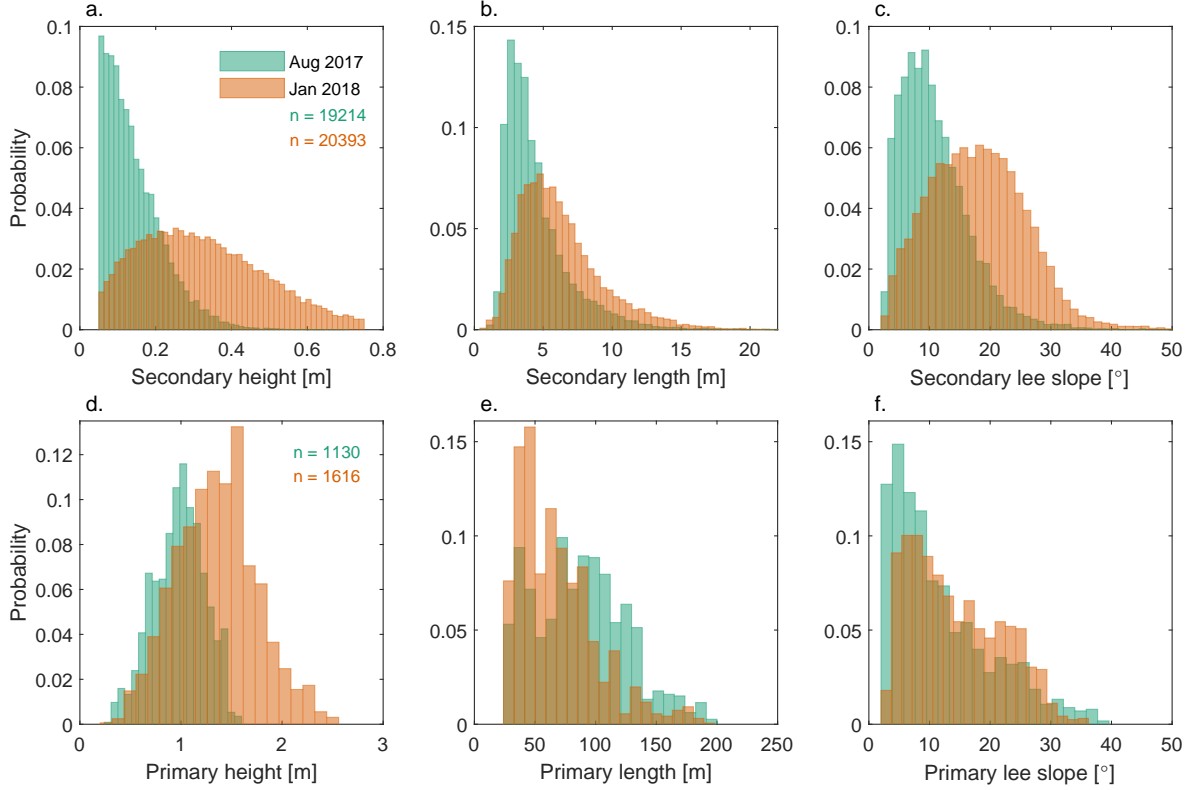

**Figure 4.** Histograms of the height, length and maximum lee side slope of secondary and primary dunes.

side slopes determine the local flow and sediment dynamics over dunes, indicate whether flow separation occurs, and determine hydraulic roughness (Kwoll et al., 2016, 2017; Lefebvre and Winter, 2016; Bradley et al., 2013).

Figure 3 shows the bedform identification based on zero-crossing, indicating that most secondary bedforms are identified.
However, Figure 4 reveals that especially during low discharge, secondary bedforms can be so small that it is difficult to clearly distinguish them from random data inaccuracies, leading to a one-sided histogram with lower values missing.

In the application of this method, a user should be careful in choosing parameters with which the tool is applied. The quality of the decomposition and with that, bedform identification and characterization, depends to some extent on the initial parameters $d_x$, the window length and on the cutoff slope. These parameters provide strong control to the user, and the parameter
settings may require tuning. Though zero-crossing is a well recognized method to identify bedforms (Van der Mark and Blom, 2007), a limitation is that, especially for raw data, random fluctuations around the zero-line can be included as bedforms, leading to a need for filtering.

For both primary and secondary dunes, a mean lee side angle and a maximum lee side angle are calculated. Usually, the lee side slope is not straight, which is relevant for lee side processes such as flow separation. For high-angle dunes, the steepest
section of the lee slope is also referred to as the slip face angle, which exerts a control over sediment avalanching (Lefebvre

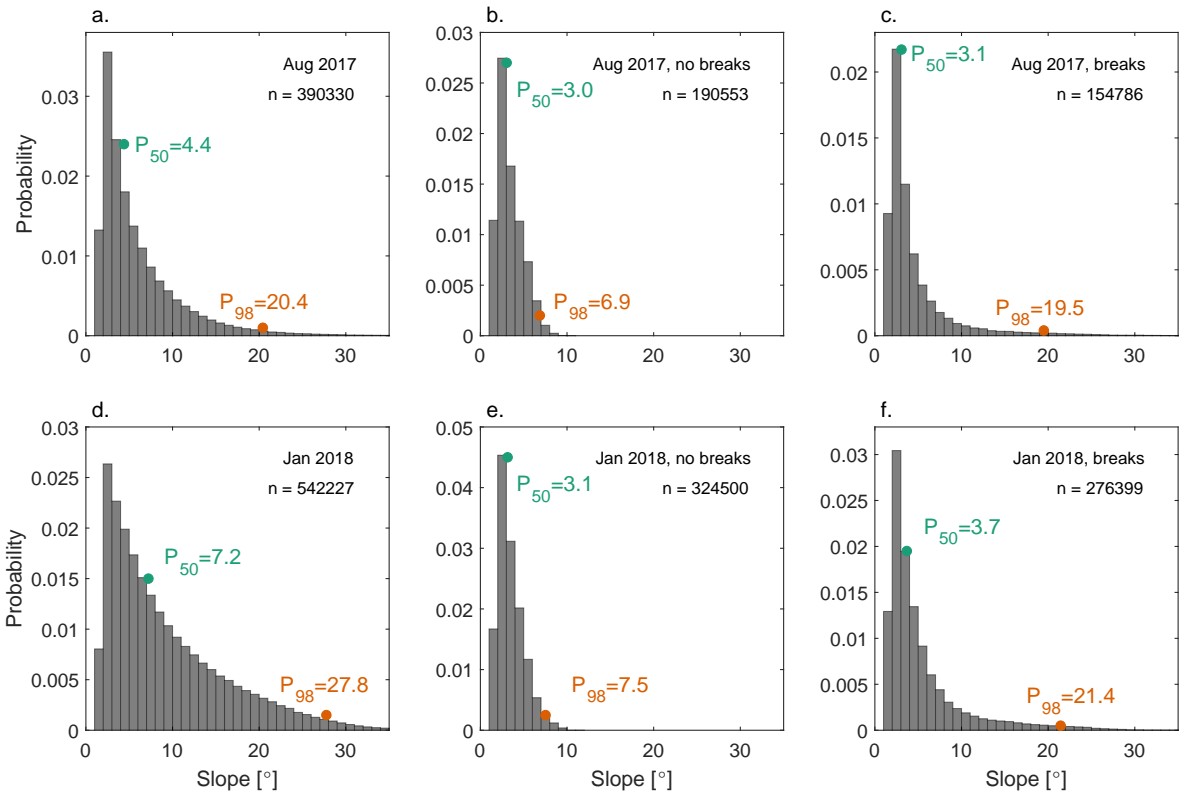

**Figure 5.** Histrograms of slope of downstream facing cells with values larger than $0.03\,\mathrm{m\,m^{-1}}$. The first column (a, c) shows the results for the original data, the second column (b, e) shows the results for the decomposed, primary morphology if no breaks are implemented and the third column (c, f) shows the results for the decomposition with the new tool, including breaks. The 50th ($P_{50}$) and 98th percentile ($P_{98}$) are displayed for each histogram.

et al., 2016; Kostaschuk and Venditti, 2019). Up- and downstream of the slip face, the lee side slope is gentler. The maximum lee side slope is determined as the maximum slope of a grid cell between the crest and the downstream trough, similar to the approach of Cisneros et al. (2020), whereas Lefebvre et al. (2016) and Van der Mark and Blom (2007) use a different approach. Lefebvre et al. (2016) defines the slip face as the part of the bedform lee which has an angle larger than 5 degrees and Van der
Mark and Blom (2007) exclude 1/6th of the lee side slope both towards the crest and towards the trough and defines the slip face based on the slope of the remaining section. The advantage of computing the maximum lee side slope of a single cell is that it is independent of the particular shape of the lee side and it avoids underestimation of the slip face. For primary dunes, the maximum slope is determined based on the decomposed signal. So for high-angle dunes, it is based on the sigmoid function fit. For secondary bedforms, the maximum slope is based on the unfiltered signal. It is important that this signal does not contain
irregularities due to measurement uncertainty, because under those circumstances, the slope of a single cell does not accurately

reflect the slope of the bedform. If this is the case, the signal should be smoothed to exclude such irregularities or otherwise, the maximum lee slope should be based on a larger number of adjacent cells.

The tool presented here is appropriate for data sets with multiple scales of bedforms, as long as these scales are sufficiently separated, and the longitudinal resolution of the data is high enough, relative to the length of the smallest bedform scale. In this study, the smallest bedform lengths are five times the longitudinal data resolution of 0.1 m. The tool is appropriate for primary dunes with steep lee side slopes. In this study, the tool is applied to a data set with subaqueous bedforms under unidirectional flow. We expect the tool can be applied to data sets from different environments, if above mentioned requirements hold and primary dune lee sides are similarly shaped. In tidal environments, primary dunes can have both an ebb and flood steep face (Lefebvre et al., 2022), for such a case we do not expect the tool to be appropriate as is, since only steep lee side slopes are approximated with the sigmoid function. For large data sets, computational time can become relevant. Application of the tool to the data sets in this study required a computational time of approximately 600 seconds without parallel processing and around 200 seconds using parallel processing. In the presented study, two bedform scales have been separated. When three or more bedform scales are present however, part of the procedure can be repeated. After separating the smallest scale bedform from the remainder, the latter signal can be subjected to the same procedure.

## 5  Conclusion

A tool is presented to decompose bed elevation data representing multiple scales for the identification and characterisation of larger primary and smaller secondary, superimposed bedforms. A LOESS algorithm was used to isolate the secondary dunes from primary dunes in between breaks downstream of steep primary lee side angles. The steep lee side slopes of primary dunes are approximated with a sigmoid function, replacing the LOESS fit at the slope. The decomposed data series are used to identify both secondary and primary dunes through zero-crossing and to measure dune properties based on filtered and unfiltered bed elevation profiles. The results show that the tool is successful in separating scales for data sets with well-defined bedform scales.

*Code and data availability.*  The MATLAB code used in this study can be accessed through https://github.com/j-zomer/BedformSeparation_Identification. The data used to make Figure 4, which includes properties of primary and secondary dunes, can be retrieved through https://doi.org/10.4121/19620093.

*Author contributions.*  Initiation and design of the study was a result of discussion between all co-authors. Judith Zomer developed the tool, about which the co-authors were regularly consulted. Judith Zomer analyzed the data and wrote the manuscript. The manuscript was reviewed and edited by all co-authors.

*Competing interests.* There are no competing interests.

*Acknowledgements.* This study is part of the research program Rivers2Morrow, which is funded by the Dutch Ministry of Infrastructure and Water Management and its executive organization Rijkswaterstaat. We thank Rijkswaterstaat-CIV for providing the data used in this study. A.J.F. Hoitink and S. Naqshband were partially funded by the Netherlands Organization for Scientific Research (NWO), within Vici project "Deltas out of shape: regime changes of sediment dynamics in tide-influenced deltas" (Grant NWO-TTW 17062). We thank Ray Kostaschuk and one anonymous reviewer for their helpful comments and suggestions.

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
