# Peer review of "A tool for determining multiscale bedform characteristics from bed elevation data"

_Earth Surface Dynamics, 2021_

## Referee Comment (RC1)

**General comments**

The authors developed a new bedform tracking tool based on natural river bathymetry data with the particular purpose of separating distinct bedform scales when several hierarchies of bedforms coexist. The introduced bedform tracking tool in this manuscript is expected to play an important role in understanding multi-scale bedform dynamics and help decode their different behaviors and potential dependencies among them. The manuscript is clearly and concisely written, and would be of great interest to readers of *Earth Surface Dynamics.* I truly support the authors' work and the manuscript is worthy of publication without any doubts. However, some minor changes need to be made before publication for clarification and presentation of the work.

**Specific comments**

1. Lines 8-10: please rework the sentence, "The approach to decompose bedforms adopted in the presented tool is particularly applicable where secondary dunes are large and thus filtering could easily lead to undesired smoothing of the primary morphology." I could fully understand what this sentence actually means after reading the manuscript. The main source of confusion is that there was no explanation about the referred filtering in this sentence. I am sure the authors meant conventional smoothing filters widely used in bedform tracking, but it is worth specifying it in the sentence or before.

2. Line 44: please distinguish other methods that apply smoothing algorithms and spectral methods. From my understanding, the main disadvantage of using spectral methods is that bedform shape needs to be pre-defined with base functions (e.g. wavelet or sinusoidal functions) and it is assumed that self-similarity of bedform shape extends across scales. This is fairly different from drawbacks of using smoothing algorithms. In this context, it is worth mentioning and citing the following papers.

   - Ganti, Vamsi, Chris Paola, and Efi Foufoula‐Georgiou. "Kinematic controls on the geometry of the preserved cross sets." *Journal of Geophysical Research: Earth Surface* 118, no. 3 (2013): 1296-1307.
   - Lee, Jiyong, Mirko Musa, and Michele Guala. "Scale‐dependent bedform migration and deformation in the physical and spectral domains." *Journal of Geophysical Research: Earth Surface* 126, no. 5 (2021): e2020JF005811.

3. Lines 45-46: I understand that the main advantage of the bedform tracking tool in this manuscript is preserved steep lee side angle of primary bedforms. But it is also important to note that the conventional smoothing filter can distort morphology of secondary bedforms.

4. Method section: there are other polynomial function based fitting algorithm like LOESS curve (e.g. Savitzky-Golay filter) as well as decaying functions like Sigmoid. I am wondering whether the authors have applied other methods for extracting primary bedforms and preserving their lee-side angle. I don't think sensitivity analysis using other algorithms is necessary in this manuscript since the introduced bedform tracking is robust and works well. In addition, there might be only small discrepancies in results obtained from using different fitting and decaying functions. But it might be worth mentioning potential candidates for the smoothing and decaying functions because this manuscript focuses on the technicality of the new bedform tracking method. This would

allow readers to know available alternatives in the fitting algorithms and replace them if needed. Addressing what considerations need to be made in selecting algorithms would also be appreciated.

5.  Method section: please add a unit of degree for the cutoff slope to give a better idea on the steepness of the slope.

6.  Figure 3: it is interesting to see asymmetric primary bedforms in the first top two panels and more symmetric primary bedforms in the bottom two panels. Any brief comments on the potential reasons would be appreciated. It seems to me this is beyond the scope of the work, so addressing this in the manuscript is not required.

7.  Lines 186-190: potential opportunities and limits of applying this method can be addressed here in detail. I suppose the introduced bedform tracking tool in this manuscript would work well in characterizing most of the riverine bedforms with unidirectional flows. However, would this method work in aeolian dunes or tide induced bedforms? What considerations need to be made before extending this method to bedforms created in other environments.

**Conclusion**
Minor revision

---

## Author Comment (AC1)

**Response to the reviews of 'Short communication: A new tool to define multiscale bedform characteristics from bed elevation data' by J.Y. Zomer et al.**

We thank the reviewers for their positive evaluation of our manuscript, and the useful suggestions for improvement. Below, we respond to the reviewer comments.

**RC1**

General comments

Many thanks for the support.

Specific comments

1. Lines 8-10: please rework the sentence, "The approach to decompose bedforms adopted in the presented tool is particularly applicable where secondary dunes are large and thus filtering could easily lead to undesired smoothing of the primary morphology." I could fully understand what this sentence actually means after reading the manuscript. The main source of confusion is that there was no explanation about the referred filtering in this sentence. I am sure the authors meant conventional smoothing filters widely used in bedform tracking, but it is worth specifying it in the sentence or before.

We have added 'using conventional continuously differentiable functions' after filtering to specify this more clearly.

2. Line 44: please distinguish other methods that apply smoothing algorithms and spectral methods. From my understanding, the main disadvantage of using spectral methods is that bedform shape needs to be pre-defined with base functions (e.g. wavelet or sinusoidal functions) and it is assumed that self-similarity of bedform shape extends across scales. This is fairly different from drawbacks of using smoothing algorithms. In this context, it is worth mentioning and citing the following papers.
   - Ganti, Vamsi, Chris Paola, and Efi Foufoula-Georgiou. "Kinematic controls on the geometry of the preserved cross sets." Journal of Geophysical Research: Earth Surface 118, no. 3 (2013): 1296-1307.
   - Lee, Jiyong, Mirko Musa, and Michele Guala. "Scale-dependent bedform migration and deformation in the physical and spectral domains." Journal of Geophysical Research: Earth Surface 126, no. 5 (2021): e2020JF005811.

To address this comment, we rewrote lines 44 to 47 to add specific references along with the alternative filtering approaches. We also explicitly mention the disadvantage of using spectral filters. In low-pass filtering, the high-frequency base functions are removed, which are indispensable when representing the asymmetric shape of high-angle dunes (Lee et al., 2021). We rewrote lines 44 to 47 to: *'Other methods apply filtering based on geostatistics (Van Dijk et al., 2008), spectral methods (Van Dijk et al., 2008, Lee et al., 2021, Cazenave et al., 2013) or spline functions following spectral analysis (Gutierrez et al., 2013, Wang et al., 2020). When using spectral filters, the reconstruction of the asymmetric shapes of high-angle dunes cannot do without high-frequency base functions, which are removed in the filtering process, in turn leading to a misrepresentation of the steep lee side slope in the lowpass filtered signal (Lee et al., 2021). Also in other filtering methods, a high degree of smoothing, which is inevitable in the presence of larger secondary bedforms, significantly affects the primary lee side slope leading to missing secondary bedforms on the lee side, or at the least significant underestimation of the lee side slopes.'*

3. Lines 45-46: I understand that the main advantage of the bedform tracking tool in this manuscript is preserved steep lee side angle of primary bedforms. But it is also important to note that the conventional smoothing filter can distort morphology of secondary bedforms.

We acknowledge the validity of this comment. We think the distortion is especially an issue if the high-pass filtered signal is used as a basis when characterizing the secondary dunes. We have added this notion in the revised manuscript.

4.  Method section: there are other polynomial function based fitting algorithm like LOESS curve (e.g. Savitzky-Golay filter) as well as decaying functions like Sigmoid. I am wondering whether the authors have applied other methods for extracting primary bedforms and preserving their lee-side angle. I don't think sensitivity analysis using other algorithms is necessary in this manuscript since the introduced bedform tracking is robust and works well. In addition, there might be only small discrepancies in results obtained from using different fitting and decaying functions. But it might be worth mentioning potential candidates for the smoothing and decaying functions because this manuscript focuses on the technicality of the new bedform tracking method. This would allow readers to know available alternatives in the fitting algorithms and replace them if needed. Addressing what considerations need to be made in selecting algorithms would also be appreciated.

We have tested various approaches to improve the fit of the primary lee side slopes, but we did not try multiple smoothing algorithms or decaying functions, similar to the sigmoid function. The LOESS algorithm was selected because it uses locally weighted regression and no predefined global function is required. Also, no information is lost at the start and at the end of a spatial series. LOESS is appropriate for smoothing highly irregular primary dune shapes in terms of lengths and heights. Other approaches might also be suitable, depending on the specific dataset and, for example, whether the datapoints are equidistant or not. The success of a smoothing algorithm may be dependent on the bathymetry to which it is applied. For the data we applied, our approach leaves little room for improvement. We have added a short comment on this topic to line 80: '*A user may choose to use an alternative smoothing algorithm to replace LOESS, or use an alternative S-shaped function to replace the sigmoid function. Examples of smoothing algorithms are the Savitzky-Golay filter or a kernel smoothing algorithm. The sigmoid curve could potentially be replaced by a hyperbolic tangent function. The LOESS algorithm was selected here because it is considered appropriate to fit primary dunes with irregular shapes in terms of heights and lengths and where deformation of bedforms is significant (Ganti et al., 2013). Also, no information is lost at the start and at the end of a spatial series. Practical considerations in selection of an algorithms could be computational time and whether data points are equidistant or not.*'

5.  Method section: please add a unit of degree for the cutoff slope to give a better idea on the steepness of the slope.

    Added to line 85 (step 2).

6.  Figure 3: it is interesting to see asymmetric primary bedforms in the first top two panels and more symmetric primary bedforms in the bottom two panels. Any brief comments on the potential reasons would be appreciated. It seems to me this is beyond the scope of the work, so addressing this in the manuscript is not required.

Indeed, this is especially visible in the lower panel. Primary dunes seem more regularly spaced and have parallel, but curved crestlines. In general, in the Dutch river Waal, primary dunes become higher, steeper, and shorter during high discharge.

7.  Lines 186-190: potential opportunities and limits of applying this method can be addressed here in detail. I suppose the introduced bedform tracking tool in this manuscript would work well in characterizing most of the riverine bedforms with unidirectional flows. However, would this method work in aeolian dunes or tide induced bedforms? What considerations need to be made before extending this method to bedforms created in other environments.

In the current approach, only steep lee sides can be fitted with the sigmoid function. In tidal areas, dunes can have steep slopes on both sides (Lefebvre et al., 2022). Under such circumstances, our approach needs to be extended. This is added to the Discussion section. Other considerations are also mentioned in Lines 186 to 190.

**References**

Lefebvre, A., Herrling, G., Becker, M., Zorndt, A., Krämer, K. & Winter, C. (2022) Morphology of estuarine bedforms, Weser Estuary, Germany. Earth Surface Processes and Landforms, 47(1), 242– 256. Available from: https://doi.org/10.1002/esp.5243

Lee, J., Musa, M., and Guala, M. (2021) Scale-dependent bedform migration and deformation in the physical and spectral domains, Journal of Geophysical Research: Earth Surface, 126, e2020JF005 811

Ganti, Vamsi, Chris Paola, and Efi Foufoula-Georgiou. "Kinematic controls on the geometry of the preserved cross sets." Journal of Geophysical Research: Earth Surface 118, no. 3 (2013): 1296-1307.

**RC2**

General comments

Many thanks for the support.

Specific comments

line 56:The LOESS method, as I understand it, is used on bed profiles so prior to the 'first step' profiles must be selected. There seems to be little or no agreement in the literature on how or where profiles are selected - however this is beyond the scope of the present paper, which assumes profiles are already made. So there should be an opening sentence(s) such that pre-selected profiles are required and how these were collected in this study.

> Here, we employ curvilinear grids with a spatial resolution of 0.1m in the longitudinal direction and 1.0m in the lateral (cross-river) direction. We analyse nearly all longitudinal profiles across the grid, to capture all transverse variability. We only exclude areas near the banks where the river bed is heavily influenced by groynes (scours). Of course, a user may choose to make a selection of profiles. To clarify the above, we have added '*BEPs analyzed here include every longitudinal grid line between -81 m and 82 m with respect to the river's central axis, which is the region unaffected by scours induced by river groynes*.' to section 2.3 Data description.

**Lines 121-122**. Clarification is required at the end of this paragraph. My understanding is that a sigmoid function is used to isolate the steep downstream-facing slope of primary dunes, as shown on Figure 1. The sigmoid function is not used on secondary dunes however so the sigmoid slope is an additional parameter.

> We added clarification at the end of the paragraph to explain how the maximum lee side slope is defined for the primary dunes. '*The maximum lee slope is defined as the maximum slope of a grid cell between the crest and the downstream trough. If the primary lee side is fitted using the sigmoid function, the maximum slope is determined based on values corresponding to the fitted function only*.' is added after line 122.

**Line 151**. I think it might be useful to have a sentence here on the position of the 2 sets of measurements on the hydrograph - e.g. on the falling or rising limb, at the peak etc. in order to better understand the data. This paper may be helpful: Bradley, R. W., & Venditti, J. G. (2021). Mechanisms of dune growth and decay in rivers. Geophysical Research Letters, 48, e2021GL094572. https://doi.org/10.1029/2021GL094572

> We added information about the timing of the two campaigns in the hydrograph: '*The campaign in August followed an extended period of low discharge. Secondary bedforms were relatively small and only partly cover the primary dunes. The campaign in January 2018 took place during the rising limb just before a peak discharge, which succeeded several similar peaks in previous months. During high discharges, primary dunes are shorter, and large secondary bedforms cover a large part of bed. In the southern river section secondary bedforms are dominant whereas primary dunes nearly disappear. In the two cases presented here, secondary bedforms are not part of a decaying process, where they cannibalize larger primary dunes (Bradley and Venditti, 2021). Two scales coexist during low flow and a rising hydrographic limb, similar to what has been observed in the Waal river by Zomer et al. (2021).*'

> This information may be particularly relevant for the campaign in August. Considering that the discharge had been low for about 6 months prior to the campaign, we expect that both dune scales have had sufficient time to adapt to the lower discharge and are in a (near-)equilibrium state.

**Line 191** (also see comments on lines 121-122): A short paragraph should be added to the end of the Discussion clarifying 'lee side slope', with particular reference to Figure 4. The plots of the primary lee slope on Figure 4 are the 'sigmoid' fit, which is equivalent to the 'slipface' as defined by Kostaschuk, R.A., and Venditti, J.G., 2019, Why do large, deep rivers have low-angle dune beds?: Geology, v. 47, p. 919–922, https://doi.org/10.1130/ G46460.1. I think this is important because the slipface angle is a critical diagnostic parameter for lee side processes. Also, the secondary lee side slope seems to be the 'mean leeside' angle from the crest to the downstream trough, which is a different measurement that the sigmoid/slipface angle for the primary dunes.

> In the Methods section, we explain how the mean lee side slope and maximum lee side slope (which is closer to the slip face angle) are calculated. We added the following paragraph in the discussion section: '*For both primary and secondary dunes, a mean lee side angle and a maximum lee side angle are calculated. Usually, the lee side slope is not straight, which is relevant for lee side processes such as flow separation. For high-angle dunes, the steepest section of the lee slope is also referred to as the slip face angle, which exerts a control over sediment avalanching (Lefebvre et al., 2016, Kostaschuk and Venditti, 2019). Up- and downstream of the slip face, the lee side slope is gentler. The maximum lee side slope is determined as the maximum slope of a grid cell between the crest and the downstream trough, similar to the approach of Cisneros et al. (2020), whereas Lefebvre et al. (2016) and Van der Mark and Blom (2007) use a different approach. Lefebvre et al. (2016) defines the slip face as the part of the bedform lee which has an angle larger than 5 degrees. Van der Mark and Blom (2007) exclude $1/6^{th}$ of the lee side slope both towards the crest and towards the trough, and defines the slip face based on the slope of the remaining section. The advantage of computing the maximum lee side slope of a single cell is that it is independent of the particular shape of the lee side, and it avoids underestimation of the slip face. For primary dunes, the maximum slope is determined based on the decomposed signal. So, for high-angle dunes, it is based on the sigmoid function fit. For secondary bedforms, the maximum slope is based on the unfiltered signal. It is important that this signal does not contain irregularities due to measurement uncertainty, because under those circumstances, the slope of a single cell does not accurately reflect the slope of the bedform. If this is the case, the signal should be smoothed to exclude such irregularities or otherwise, the maximum lee slope should be based on a larger number of adjacent cells.*'

**Figure 4:** Add the number of observations N for each histogram. Also, 'lee slope' should be defined clearly in the text and/or the caption, as noted with reference to line 191 above.

> We changed the caption of figure 4 to: '*Histograms of the height, length and maximum lee side slope of secondary and primary dunes.*' We further added the number of observations to the plots. The lee side slope is also more elaborately discussed in the Discussion section, see also the response to the comment that refers to line 191.

**Figure 5:** The number of observations N should be included on each histogram. Also define D50 (median?) and D98 (coarse percentile?) in the caption.

> The D50 and D98 should be P50 and P98. We changed this in the figure, and the following sentence is added to the caption of Figure 5: '*The 50th (P50) and 98th percentile (P98) are displayed for each histogram.*' We also included values outside the bounds as specified in section 2.3 entitled Data description.

**Supplementary material:** The data for Figure 4 should be provided so that other researchers can use it in their studies.

> We will provide the data for Figure 4 (characteristics of identified primary and secondary bedforms).

**Technical corrections**

**Abstract**

Lines 1-2: delete "focused on" and replace with 'of'

>      Changed accordingly.

Lines 4-5: delete "multiple dune scales" and replace with 'of large primary and smaller, superimposed, secondary dunes'; delete "based on" and replace with 'using'

>      Changed accordingly.

Line 5: delete "dune" (in "Steep dune lee side); delete "are accounted for" and replace with 'of primary dunes are identified'

>      We changed the sentence to '*Steep lee side slopes of primary dunes are preserved by implementing ... '*

Line 6: delete ", often occurring" and "of dunes" (the latter at the end of the sentence)

>      Changed accordingly.

Line 8: delete the ", and the relevant" and replace with 'and morphological'

>      Changed accordingly.

Lines 8-9: delete "decompose bedforms adopted in the presented tool" with 'bedform decomposition presented herein'

>      Changed accordingly.

Lines 10-11: delete "the decomposition and identification are successful, as the lee side slopes are better preserved" and replace with 'it successfully decomposes bathymetric data, identifies primary and secondary dunes, and preserves steeper lee side slopes of primary dunes.'

>      Changed accordingly.

**Introduction**

Line 15: delete "More, in general" and replace with 'Moreover'

>      Changed accordingly.

Line 16: delete ", associated with this, "

>      Changed accordingly.

Line 28: delete "those" and replace with 'these'.

>      Changed accordingly.

Line 44: delete "These" and replace with 'However, these'

>      Changed accordingly.

Line 45: delete "In case" and replace with 'In the case'

>      Changed accordingly.

Line 50: add the following sentence after the sentence that ends with "including the primary dunes": 'LOESS regression is a nonparametric technique that uses local weighted regression to fit a smooth curve through points in a scatter plot.'

> Changed accordingly.

Line 50: delete "Different" and replace with 'LOESS differs'

> We did not adopt the proposed change (replace 'Different' with 'LOESS differs'), as the sentence is about the implementation of breaks in the LOESS curve. A LOESS curve without breaks would be continuously differentiable, similar to other approaches. We replaced 'Different' with 'The approach differs'

Line 51: add 'in that' in front of "no continuously"

> Changed accordingly.

Line 51: add 'rather' in between the words "but" and " breaks"

> Changed accordingly.

Line 52: delete "implemented" and replace with 'used'

> Changed accordingly.

Line 52: delete "feature" and replace with 'contain'

> Changed accordingly.

**Methods**

Line 62: how is the "user-defined" determined?

> The default value is specified in line 87. A break should be implemented when the slope of the LOESS curve at the lee side deviates (is smaller than) the slope of the primary lee side in the original signal. The value also depends on the extent of smoothing, which is captured by $d_x$, the half-span of the smoother.

Line 83: delete "explained below"

> Changed accordingly.

Line 90: should "lower" be 'larger'?

> 'lower' is correct. This is done to fit the sigmoid function to the slip face and exclude the trough.

Line 93: 'central' should be defined. e.g., 'central 50%', or 10% etc.

> What is removed is specified in the following sentences. To clarify this, we replaced '... decomposition. The ...' with '... decomposition: the ....'.

Line 101: delete "are" and replace with 'is'

> Changed accordingly.

Line 109: delete "Subsequently, bedform properties are being determined. Properties" and replace with 'Secondary bedform properties'

> Changed accordingly.

Lines 108-116: This paragraph could be added to the previous paragraph since both paragraphs are about secondary bedforms.

Changed accordingly.

Line 110: delete "trough" and replace with 'troughs'

We did not adopt this change, it would result in the sentence 'crest and troughs locations', which seems incorrect.

Line 120: delete ". Again, we iterate" with 'iterating once'

We replaced 'Again we iterate' with 'Primary dunes are identified, iterating once'.

Line 120: does "smaller than 0.25 m" refer to height or length?

This refers to height. We replace 'If primary dunes are smaller than 0.25 m' with 'If a primary dune height is smaller than 0.25 m'.

Line 123: delete "identified"

Changed accordingly.

Line 124: delete " e.g. as a result of" and replace with 'such as

Changed accordingly.

Lines 125-136: These filters are subjective but I see no way of getting around this. It might be useful to add 'and site-specific' after "user-defined"

Changed accordingly.

Lines 133: delete "then" and replace with 'than'

Changed accordingly.

Line 143: delete "trough" and replace with 'through'

Changed accordingly.

Lines 146-147: : I don't think 'dx' has been defined previously so it should be done so here.

$d_x$ has been defined in line 75.

**Results**

Line 150: delete "Especially in" and replace with 'In'.

Changed accordingly.

Line 151: delete the comma after "section"

Changed accordingly.

Line 154: delete "tops" and replace with 'crests'

Changed accordingly.

Line 162: delete "substantially"

Changed accordingly.

**Discussion**

Line 170: delete the period at the end of the sentence

>Changed accordingly.

Line 171: delete "A" at the beginning of the first sentence and replace with 'because a' - merging this sentence with the sentence on line 170

>Changed accordingly.

Line 172: delete "indicates" in the second sentence and replace with 'shows'

>Changed accordingly.

Line 174: delete "in many studies" and replace with 'because'

>Changed accordingly.

Line 177: delete "illustrating" and replace with 'indicating'

>Changed accordingly.

Line 178: add 'However' to the beginning of this sentence

>Changed accordingly.

Lines 182-183: delete ", and in using the tool, the" and replace with 'and'

>Changed accordingly.

Line 184: delete "downside" and replace with 'limitation

>Changed accordingly.

Line 188: the statement "data resolution" needs clarification, e.g., is this the horizontal or vertical resolution

>We replaced '... *and the resolution of the data is high enough, relative to the length of the smallest bedform scale. In this study, the smallest bedforms are five times the data resolution'* with '*... and the longitudinal resolution of the data is high enough, relative to the length of the smallest bedform scale. In this study, the smallest bedform lengths are five times the longitudinal data resolution of 0.1 m'*.

Line 188: delete " symmetrical and asymmetrical" and replace with 'primary' - I think the authors are referring to primary, not secondary, dunes here.

>Changed accordingly.

Line 189: what is meant by "steep lee side slopes" here? Is this referring to the sigmoid fit of primary dunes?

>The tool is appropriate for application to bathymetry containing steep primary lee side slopes, because those are preserved by implementing breaks and introducing the sigmoid fit.

Line 189: delete ",potentially,"

Changed accordingly.

**Conclusions**

Line 193: delete "two scales of" and replace with 'large primary and smaller secondary, superimposed,'

Changed accordingly.

Line 193: delete "the highest scale" and replace with 'primary

See the response to the comment regarding line 194.

Lline 194: delete "primary' in "steep primary lee side angles"

See the response to the comment regarding line 194.

Line 194: delete "immediately downstream of the breaks" and replace with 'of primary dunes

We replaced '*A LOESS algorithm was used to isolate the highest scale bedforms in between breaks downstream of steep primary lee side angles. The steep lee side slopes immediately downstream of the breaks are approximated with a sigmoid function, replacing the LOESS fit at the slope*' with '*A LOESS algorithm was used to isolate the secondary dunes from primary dunes in between breaks downstream of steep primary lee side angles. The steep lee side slopes of primary dunes are approximated with a sigmoid function, replacing the LOESS fit at the slope.*'

---

## Author Response (AR2)

Dear Niels Hovius and Wolfgang Schwanghart,

Thank you for accepting our manuscript for publication in ESurf. We have made the final changes that were requested, as outlined below.

With kind regards,

Judith Zomer

Title: The title has been changed to 'A tool for determining multiscale bedform characteristics from bed elevation data'.

Line 80: Has been changed to: 'The weights $w_j$ are defined by a tricube weight function as...'

Line 224f: The following statement has been added to line 230: '*For large data sets, computational time can become relevant. Application of the tool to the data sets in this study required a computational time of approximately 600 seconds without parallel processing and around 200 seconds using parallel processing.*'

Fig 1-3: '*Flow direction*' has been added to the arrow indicating the flow direction.

Fig 2-5: Letters have been added to the figures and the corresponding captions have been changed as well as the text where necessary to include the letters when referring to a specific panel.

We also changed Matlab to MATLAB throughout the manuscript.

Concerning the code on github: The two proposed changes will be in the next release of the software.